# β Grain Size Inhomogeneity of Large Scale Ti-5Al-5V-5Mo-3Cr Alloy Bulk after Multi-Cycle and Multi-Axial Forging in α + β Field

**DOI:** 10.3390/ma16041692

**Published:** 2023-02-17

**Authors:** Dongyang Qin, Huifang Liu, Yulong Li

**Affiliations:** 1School of Aeronautics, Northwestern Polytechnical University, Xi’an 710072, China; 2Department of Engineering Science, University of Oxford, Oxford OX1 3JP, UK; 3School of Civil Aviation, Northwestern Polytechnical University, Xi’an 710072, China

**Keywords:** titanium alloys, sub-transus forging, dynamic recrystallization, abnormal grain growth

## Abstract

In order to fabricate homogeneous large-scale Ti-5Al-5V-5Mo-3Cr (Ti-5553) alloy bulk with fine and equiaxial β grain, we performed a series of multi-axial α + β field forging with 62 forging cycles on the large-scale Ti-5553 billet by using 12.5 MN high-speed hydraulic press. The β-annealed microstructure was the starting microstructure of the billet. After the 6th forging cycle, β grain deformed dramatically, and the grain-boundary network developed within the irregular β grain. As the forging cycle increased to 44, the volume fraction of the fine and equiaxial β grain that is less than 20 μm, which is caused by dynamic recrystallization, increased gradually. However, the incomplete dynamic recrystallization region within the original β grain could not be eliminated. As the forging cycle further increased, the volume fraction of the fine and equiaxial β grain did not increase. In contrast, the abnormal grain growth of the β phase occurred during 50th~62nd forging cycle. Here, we attribute the formation of the incomplete dynamic recrystallization region and the abnormal grain growth of the β phase to the high deformation rate of the α + β forging. The refining behavior of β grain and the abnormal coursing β grain, which is found during the multi-cycle multi-axial forging of large-scale Ti-5553 alloy billet, are seldom reported in the isothermal compression of small-scale Ti-5553 alloy specimen. The findings of the paper are instructive for improving the sub-transus forging strategy that is used to fabricate the large-scale homogeneity Ti-5553 alloy billet with fine and equiaxial β grain.

## 1. Introduction

Near-β titanium alloys are widely used in the aviation industry because of their high strength-to-weight ratio and excellent corrosion resistance [1]. Based on the grain size of β grain, the microstructure of high-strength near-β titanium alloys could be classified into two categories, which are β-annealed microstructure and bimodal microstructure [2,3]. β-annealed microstructure processes coarse equiaxial prior β grains, and the average grain size of prior β grains is approximately 200 μm [3]. Microstructure for the forging of airframe components is seldom β-annealed microstructure, because the coarse prior β grain is considerably harmful to the fatigue limit and the ductility of the parent alloy. In contrast, bimodal microstructure processes fine and equiaxial prior β grains, and the average grain size of prior β grains is less than 20 μm [4,5]. Bimodal near-β titanium alloy demonstrates balanced mechanical properties in strength, ductility, fatigue limit, and fracture toughness. Consequently, bimodal microstructure should be the optimum microstructure for aircraft forgings that are fabricated by near-β titanium alloys.

Ti-5Al-5V-5Mo-3Cr (Ti-5553) alloy, which is a relatively new near-β titanium alloy, exhibits several unique merits such as improved hardenability and better compositional homogeneity in melting as compared to other titanium alloys [6,7]. Owing to the attainable ultra-high yield strength shown in Table 1, this new alloy can be sued in thick section forging for high-strength airframe components, e.g., landing gear, flap tracks, etc [1]. Both Boeing and Airbus are evaluating this alloy for application in structural components of future aircraft [1,8]. Recently, the mechanical properties of bimodal Ti-5553 alloy have been extensively investigated. Several researchers have found that bimodal Ti-5553 alloy indeed processes the excellent combination of strength, fatigue limit and ductility, and the receivable fracture toughness [9,10,11]. Since the β grain of Ti-5553 alloy ingot is considerably coarse, which might achieve a millimeter level, the fine and equiaxial β grain of bimodal Ti-5553 originates from the deliberative forging in α + β field. The refining mechanism of β grain is dominated by dynamic recrystallization [4]. Although the effect of α + β forging parameters, including forging temperature region and critical forging strain, on dynamic recrystallization of Ti-5553 alloy has been explored by performing the isothermal compression test of the small-scale Ti-5553 alloy cylinder [2,4,5,8,12], little work has been done on the role of forging parameters in the fabrication of large-scale Ti-5553 alloy billet. It should be pointed out that the preparation of the large-scale Ti-5553 alloy billet with fine and equiaxial β grain should be the base of the fabrication of airframe components with bimodal microstructure. In this paper, multi-cycle and multi-axial forging in the α + β field is conducted on the large-scale Ti-5553 alloy billet. The aim of the work is to explore the forging strategy that is suitable for the fabrication of large-scale homogeneous Ti-5553 alloy billet with fine and equiaxial β grain.

## 2. Materials and Methods

### 2.1. Melting and β-Field Forging

The Ti-5553 alloy ingot with the dimension of Φ 260 mm × 840 mm was melted by a consumable electrode vacuum furnace with a maximum melting weight of 1000 Kg. Substances used for melting the ingot of Ti-5553 alloy were supplied by Northwestern Institute for Non-ferrous Metal Research. Table 2 shows the chemical composition of the ingot. The β-transus temperature of the ingot was 825 °C.

The aim of the β-field forging is to eliminate the as-cast microstructure of Ti-5553 alloy ingot. The β-field forging was conducted on 12.5 MN high-speed hydraulic press. Before starting each forging cycle, the billet (ingot) was kept in the pre-heated furnace for 180 min. The temperature of the first cycle β-field forging, the second cycle β-field forging and the third cycle β-field forging were 1180 °C, 1080 °C and 900 °C, respectively. Figure 1 shows the process flow of β field forging. The forging strain rate of the billet was in the range of 5/s to 10/s. After finishing each forging cycle, the billet was quenched into the water tank with the dimension of 2500 mm × 1500 mm × 1200 mm. The final dimension of the Ti-5553 alloy billet was 200 mm × 200 mm × 900 mm.

### 2.2. Ex Situ Investigation on Microstructure of the Billet before Forging in α + β Field

#### 2.2.1. Necessity of Ex situ Investigation

In the present work, length, width, and height of the Ti-5553 billet for α + β field are 200 mm, 200 mm, and 400 mm. Before starting the α + β forging, the billet was annealed at 900 °C for 120 min, and was quenched into the water tank, and was then annealed at 785 °C for 120 min. It is hard to retain the microstructure of the billet before the α + β forging to room temperature, because the length, width, and height of the billet are larger than the maximum quenching diameter of Ti-5553 alloy [2]. Therefore, the ex situ investigation on the microstructure of large-scale Ti-5553 alloy billet before forging in α + β field was conducted.

#### 2.2.2. The Source of Metallographic Samples

The slice with the dimension of 200 mm × 200 mm × 12 mm was machined from the billet forged in the β-field, and the cylinder with the dimension of Φ 10 mm × 10 mm was cut from the Ti-5553 alloy slice. To explore the microstructure of the billet before α + β field forging, an annealing treatment was conducted on the cylinder. The cylinder was first annealed at 900 °C for 120 min following water-quenching, and was then annealed at 785 °C for 120 min following water-quenching. The microstructure of the cylinder was examined by Zeiss Axio Vert A1 optical microscope. The observed surface of the metallographic specimen was the top surface of the cylinder. The normal of the observed surface was parallel to the Z direction of the billet.

### 2.3. Multi-Cycle and Multi-Axial Forging in α + β Field

#### 2.3.1. Forging Strategy

The aim of the α + β field forging is to refine the β grain of the large-scale billet to the grain-size range from 5 μm to 15 μm. α + β field forging of the billet was conducted on a 12.5 MN high-speed hydraulic press. The α + β forging temperature was 785 °C. Before starting each forging cycle, the billet was kept in the pre-heated furnace for 120 min. The multi-axial forging strategy for each cycle was shown in Figure 2 [13], after which the height-to-length ratio of the square billet was approximately 2. The forging strain rate of the billet was in the range of 5/s to 10/s. After each forging cycle, the square billet was air-cooled to room temperature, and the crack on the surface of the billet was removed by an angle grinder. In total, 62 forging cycles were performed.

#### 2.3.2. The Source of Metallographic Samples

After the 6th forging cycle, the 14th forging cycle, the 22nd forging cycle, the 38th forging cycle, the 44th forging cycle, the 50th forging cycle, the 56th forging cycle, and the 62nd forging cycle, a slice with a thickness of 20 mm was cut from the end of the square billet. The cylinder with the dimension of Φ 10 mm × 10 mm was cut in the center of each slice, in order to analyze the β grain size of the billet.

### 2.4. Characterizing Grain Boundary of β Grain

Figure 3a shows the optical micrograph of the alloy in forged condition (the 14th cycle). The bright contrast region is the primary α phase, while the dark region is the transformed-β microstructure that consists of β phase and secondary α phase. It is obvious the grain boundary of the prior β grain could not be seen, indicating that the size and the morphology of β grain could not be investigated by the direct microstructure observation on the forged billet. Figure 3b shows the optical micrograph of the alloy in solution-treated condition (785 °C for 1 h, following water-quenching). Although the secondary α phase disappears, the grain boundary of the prior β grain could not be seen clearly.

In our previous work, we found that after the 5 min aging at 700 °C the α precipitation forms on the β grain boundary and does not form inside the β grain [2]. Microstructure analysis reveals that the grain boundary α precipitation could be preferentially etched during etching [2]. Figure 3c shows the optical micrograph of the solution-treated billet that is aged at 700 °C for 5 min, following water-quenching. The β grain boundary could be clearly seen by using the optical microscope.

In order to investigate the role of the forging cycle in the β-grain size of Ti-5553 alloy billet and to characterize the grain boundary of β phase, the solution-ageing treatment was performed on the Φ 10 mm × 10 mm cylinder. The solution was carried out at 785 °C for 1 h following water-quenching, and the short-time ageing was carried out at 700 °C for 5 min following water-quenching. The microstructure of the Ti-5553 alloy cylinder was examined by Zeiss Axio Vert A1 optical microscope. The observed surface of the metallographic specimen was the top surface of the cylinder. The normal of the observed surface was parallel to the Z direction of the billet.

### 2.5. Isothermal Compression Test

By using the Gleeble-3800 testing machine, isothermal compression test was conducted on the alloy after the 44th cycle forging and the 62nd cycle forging. The cylinder with the dimension of Φ 8 mm × 12 mm was machined from the slices that were cut from the billet after the 44th cycle forging and the 62nd cycle forging, respectively. The K-type thermocouple was welded on the mid-height side of the specimen to monitor the temperature. To decrease the friction between the surface of the compression plate and the top surfaces of the Ti-5553 alloy cylinder, the graphite grease and the tantalum film were used. Under the clamping load of 1000 N, the specimen was heated to the target temperature at the heating rate of 6 °C/s. The specimens were compressed at 785 °C. Once at the target temperature, the specimen was held for 180 s to achieve thermal equilibrium. The strain rate of the isothermal compression test was 0.1/s. The isothermal compression strain was 0.69. After finishing the deformation, the specimen was air-cooled to room temperature. All the compressed samples were annealed at 785 °C for 1 h following water-quenching, and were then annealed at 700 °C for 5 min following water-quenching. The metallographic preparation of the heat-treated sample was carried out by sectioning it in half, embedding it in the resin, and grinding-polishing-etching techniques. The microstructure in the center of the specimen was investigated by Zeiss Axio Vert A1 optical microscope.

## 3. Results

### 3.1. Microstructure of Ti-5553 Alloy Billet before Forging in α + β Field

Figure 4 displays the optical images of the Ti-5553 alloy cylinder that was solutionized at 900 °C for 120 min following water-quenching and was aged at 785 °C for 120 min following water-quenching. Figure 4a shows the low-magnification optical image of the billet. The prior β grain is equiaxial, and most of the prior β grains are in the magnitude of 10^2^ μm. Figure 4b shows the high-magnification optical image of the billet. Grain boundary α (α_GB_) has precipitated on the grain boundary of prior β grain, and intragranular α (α_Intra_) has precipitated inside the prior β grain. Some prior β grains process the high volume fraction of α_Intra_, while the volume fraction of α_Intra_ is considerably low in the other prior β grains. In addition, the precipitation-free zone could be found in the grain boundary region. In summary, we conclude that the microstructure of the cylinder should be the typical β-annealed microstructure.

In the present work, the ex situ heat-treatment was conducted on the Φ 10 mm × 10 mm Ti-5553 alloy cylinder that was machined from the large-scale Ti-5553 alloy billet forged in β field, and the heat-treatment parameters of the cylinder were the same as those of the large-scale Ti-5553 alloy billet. Although the heating rate of the cylinder is different than that of the large-scale billet, it hardly affects the size of the prior β grain and the volume fraction of α_Intra_. As a result, the microstructure of the large-scale Ti-5553 alloy billet should also be the lamellar microstructure before starting the multi-cycle and multi-axial forging in α + β field.

### 3.2. Gradual β-Grain Refinement during the 6th~44th Cycle α + β Forging

#### 3.2.1. The 6th Cycle

Figure 5a shows the low-magnification optical image of the large-scale Ti-5553 alloy billet after the 6th cycle multi-axial forging. It is evident that the equiaxial β grain has disappeared. Figure 5b shows the high-magnification optical image recorded in the dark region of Figure 5a. β grain size has been refined, and the grain-boundary network has developed. Specially, a few ultra-fine β grains that are less than 20 μm could be found. This finding indicates that several β grains could be refined after the 6th cycle multi-axial forging cycle. Figure 5c shows the high-magnification optical image recorded in the bright region of Figure 5a. The bright region should be the deformed prior β grain with irregular morphology, in which the grain-boundary network has not developed yet. As mentioned above, the β grain size is in the magnitude of 10^2^μm before the α + β forging (Figure 4a). However, the grain size of the coarser β grain reaches 400 μm, while the finer β grain is only 100 μm. Compared with the finer β grain, it is hard to refine the coarser β grain, because the refinement of the coarser β grain requires larger thermal plastic deformation from the grain-boundary network. Here, we attribute the retaining of irregular β grain to the insufficient thermal plastic deformation within the coarser β grains.

#### 3.2.2. The 14th Cycle

Figure 5d shows the low-magnification optical image of the large-scale Ti-5553 alloy billet after the 14th cycle multi-axial forging. Compared with Figure 5a, we find that the volume fraction of the dark region increases evidently. Figure 5e displays the high-magnification optical image recorded in the dark region. The complete recrystallization region marked by the yellow circle and the incomplete recrystallization region marked by the yellow rectangle could be observed. The complete recrystallization region consists of the fine and equiaxial β grain, and α phase laminate does not precipitate within the recrystallized β grain. It is possible that the complete recrystallization region might evolve from the grain-boundary network within the β grain (Figure 5b). The domain of the incomplete recrystallization region is less than 100 μm, in which the grain-boundary network could be found. Compared with the complete recrystallization region, we find that α laminate has precipitated within the β grain of the incomplete recrystallization region. Figure 5f displays the high-magnification optical image that is recorded in the bright region of Figure 5d. The bright region is the deformed β grain that only processes the low-density grain-boundary network. These findings suggest that the original coarser β grain (Figure 4b) could not be completely refined after the 14th cycle multi-axial forging.

#### 3.2.3. The 22nd Cycle

Figure 5g shows the low-magnification optical image of the large-scale Ti-5553 alloy billet after the 22nd cycle multi-axial forging. It is evident that the irregular β grain, which has been observed in the billet after the 6th cycle forging and the billet after the 14th cycle forging, has been completely eliminated. Figure 5h displays the high-magnification optical image recorded in the bright region of Figure 5g. The bright region should be the complete recrystallization region because most of the β grains are fine and equiaxial. Figure 5i displays the high-magnification optical image recorded in the dark region of Figure 5g. Although a few recrystallized β grains could still be found, most of the β grains are not equiaxial. Our findings suggest that incomplete recrystallization is still present in the forging after the 22nd forging.

#### 3.2.4. The 38nd Cycle

Figure 5j shows the low-magnification optical image of the large-scale Ti-5553 alloy billet after the 38nd cycle multi-axial forging. Compared with the optical image of the billet after the 22nd cycle forging (Figure 5g), we find that the volume fraction of the bright region increases evidently. Figure 5k displays the high-magnification optical image recorded in the bright region of Figure 5j. The bright region should also be the complete recrystallization region. Figure 5l displays the high-magnification optical image recorded in the dark region of Figure 5j. The dark region should be the incomplete recrystallization region, because the well-recrystallized equiaxial β grain has not formed yet. The optical images of the billet after the 38nd cycle multi-axial forging suggest that the volume fraction of the complete recrystallization region increases with the forging cycle.

#### 3.2.5. The 44th Cycle

Figure 5m shows the low-magnification optical image of the large-scale Ti-5553 alloy billet after the 38nd cycle multi-axial forging, and the high-magnification optical images for the bright region and the dark region in Figure 5m are demonstrated in Figure 5n and Figure 5o, respectively. The bright region is the complete recrystallization region, while the dark region is the incomplete recrystallization region. Compared with the microstructure of the billet after the 38th forging cycle, we find the volume fraction of the complete recrystallization region increases, which has reached 75%. However, the incomplete recrystallization region has not been eliminated after the 44th cycle multi-axial forging.

#### 3.2.6. Microstructure of Incomplete Recrystallization Region

Figure 6 shows the SEM images recorded in the incomplete recrystallization region of the Ti-5553 alloy billet after the 14th forging cycle, the 22nd forging cycle, the 38th forging cycle, and the 44th forging cycle. The grain boundary of the β phase is the main grain-boundary network, and the volume fraction of the recrystallized β grain is much lower than that of the complete recrystallization region. Intragranular α precipitation could be found in the uncrystallized β phase. The formation of intragranular α particles contributes to the dark contrast of the incomplete recrystallization region that is shown in the optical image.

### 3.3. Eliminating the Incomplete Recrystallization Region

Although 44 cycles of multi-axial forging has been conducted, the incomplete recrystallization region cannot be eliminated. As a result, the isothermal compression test was performed to verify the forging strategy. Figure 7 shows the microstructure of the alloy after the isothermal compression and the solution-ageing treatment. Comparing with Figure 5o, we find that the incomplete recrystallization region has disappeared and the β grain of alloy is fine and equiaxial. It is possible that the incomplete recrystallization region of the β phase would be eliminated by the increase of the forging cycle.

### 3.4. Abnormal β-Grain Growth during the 50th~62nd Cycle α + β Forging

#### 3.4.1. The 50th Cycle

Figure 8a shows the low-magnification optical image of the large-scale Ti-5553 alloy billet after the 50th cycle multi-axial forging. The macrostructure is homogeneous, and the coarse β grain, which is observed in the billet after the 6th cycle forging and the billet after the 14th cycle forging, could not be observed. Figure 8b displays the high-magnification optical image of the alloy. Compared with the microstructure of the alloy billet after the 44th cycle multi-axial forging, we find that the volume fraction of the incomplete recrystallization region increases obviously. As a result, the complete recrystallization region distributes among the incomplete recrystallization regions. These findings suggest that the increase of the forging cycle could not further eliminate the incomplete recrystallization region. In contrast, the excessive forging cycle might reduce the recrystallized β grain.

#### 3.4.2. The 56th Cycle

Figure 8c shows the low-magnification optical image of the large-scale Ti-5553 alloy billet after the 56th cycle multi-axial forging. The macrostructure of the alloy is homogeneous, and the coarse β grain, which is observed in the billet after the 6th cycle forging and the billet after the 14th cycle forging, has not formed yet. Figure 8d displays the high-magnification optical image of the alloy. Comparing with the microstructure of the alloy billet after the 50th cycle multi-axial α + β forging, we find that the volume fraction of the recrystallized β grain is considerably low and the β phase of the alloy mainly consists of the incomplete recrystallization region.

#### 3.4.3. The 62nd Cycle

Figure 8e shows the low-magnification optical image of the large-scale Ti-5553 alloy billet after the 62nd cycle multi-axial forging. The macrostructure of the alloy is inhomogeneous, and the coarse β grain, which is approximately 50 μm, has formed. These coarse grains are marked by yellow circles. Figure 8f displays the high-magnification optical image of the alloy. The coarse β grain is surrounded by the incomplete recrystallization region. It is possible that the coarse β grain evolved from the incomplete recrystallization region. This finding suggests that the exceptional grain growth of β grain occurs after the 62nd cycle forging.

Figure 9 shows the SEM images recorded in the incomplete recrystallization region of the Ti-5553 alloy billet after the 50th forging cycle, the 56th forging cycle and the 62nd forging cycle. The grain boundary of the β phase is the main grain-boundary network, and the volume fraction of the recrystallized β grain is considerably low. Intragranular α precipitation could also be found in the uncrystallized β phase. The formation of intragranular α particles contributes to the dark contrast of the incomplete recrystallization region that is shown in the optical image.

### 3.5. Eliminating Coarse β Grain Caused by Abnormal Grain Growth

In order to eliminate the coarse β grain, which is caused by the abnormal grain growth, of the billet after the 62nd multi-axial α + β forging, isothermal compression was conducted. Figure 10 shows the microstructure of the alloy after the isothermal compression and the solution-ageing treatment. Comparing with Figure 8f, we find that the coarse β grain has disappeared, and all the β grains are well-recrystallizaed. These findings indicate that proper thermo-mechancial processing could eliminate the coarse β grain which is induced by the abnormal grain growth effectively.

## 4. Discussion

The aim of the present work is to fabricate the homogeneous large-scale Ti-5553 alloy bulk with equiaxial β grain that is less than 20 μm by multi-cycle and multi-axial forging in α + β field. However, the results contradict the original hypothesis. First, although the coarse β grain, which is retained after the β-field forging, could be refined gradually from the 6th forging cycle to the 44th forging cycle, the incomplete recrystallization region that processes un-recrystallized β grain could not be eliminated utterly. Second, the abnormal grain growth of β grain occurs during the 50th~62nd forging cycle. These data suggest our forging strategy, including forging temperature, forging strain, or forging strain rate, might be inappropriate.

### 4.1. Forging Temperature

As pointed out by Weiss, the recommended α + β forging temperature for near-β titanium alloys is in the temperature range that is 30–50 °C below the β-transus temperature [14]. In the present work, the forging temperature of α + β field is 785 °C, which is 40 °C below the β-transus temperature of the alloy. The selection of the forging temperature is reasonably consistent with the recommendation of Weiss [14]. In addition, the result of the isothermal compression at 785 °C implies that both the incomplete recrystallization region and the coarse β grain caused by abnormal grain growth could be eliminated by the isothermal compression at 785 °C. Therefore, the forging temperature of the present work should be appropriate.

### 4.2. Forging Ratio

In our previous work, we investigated the effect of the compression strain on the dynamic recrystallization behavior for the β phase of Ti-5553 alloy during the sub-transus thermomechanical processing. Our findings suggest that in the proper forging temperature range the critical strain for the dynamic recrystallization β phase is 0.30 [7]. The mechanism for the dynamic recrystallization of the β phase includes the formation of the low- angle grain boundary that is induced by the dislocation accumulation and the transformation from the low-angle grain boundary of the β phase to the high-angle grain boundary of the β phase [7].

In the present work, the strain distribution of the large-scale Ti-5553 alloy bulk during multi-cycle and multi-axial forging is more complex than that of isothermal compression. However, the dynamic recrystallization mechanism of the β phase during the engineering forging should be the same as that of the isothermal compression, which is dominated by the formation of the new grain boundary. As mentioned above, the incomplete recrystallization region of the β phase is still present in the bulk even after the 44th cycle forging cycle. It is understandable that the accumulated forging ratio of the bulk is considerably high after the 44th cycle forging cycle, and the accumulated strain for each part of the bulk should be much higher than the critical strain for the dynamic recrystallization β phase. Therefore, we conclude that the retained incomplete recrystallization region of the large-scale Ti-5553 alloy bulk is not caused by the insufficient forging strain.

### 4.3. Deformation Rate

In the present work, the multi-cycle and multi-axial α + β field were performed at 785 °C. Before starting each forging cycle, the billet was annealed at 785 °C for 120 min. As mentioned in Figure 2, the alloy contains α laminate, and the volume fraction of the primary α laminate is approximately 18.5%. Since the forging temperature of each forging cycle is the same, the volume fraction of the primary α phase of the billet should be around 18.5% before each forging cycle. In theory, the β grain of Ti-5553 alloy could be refined by the forging in α + β field, because the primary α phase could strongly inhibit the grain growth of the β phase and is beneficial to the dynamic recrystallization of the β phase. However, as mentioned in Section 2.2.2, the forging strain rate of the large-scale billet was in the range of 5/s to 10/s. The selection of the forging strain rate might be inappropriate.

First, the high forging strain rate tends to result in a drastic temperate rise in each α + β forging cycle. It is reported that the temperature rise of β titanium alloys might be higher than 45 °C during the α + β isothermal compression as the compression strain rate reaches 10/s [15]. As a result, the temperature of the billet might be close to the β-transus temperature of Ti-5553 alloy, which is 825 °C. Our previous work reveals that the β grain size of Ti-5553 alloy grows rapidly in the temperature range that is 15–25 °C lower than the β-transus temperature [8]. Therefore, it is possible that the high deformation rate might lead to the drastic β grain growth during the multi-axial and multi-cycle forging at 785 °C.

Second, the drastic temperate rise during α + β forging might lead to the dynamic α→β transformation of the billet. Li et al. reported the dynamic α→β transformation of Ti-5Al-5V-5Mo-1Cr-1Fe β titanium alloy, the chemical composition of which is considerably similar to that of the Ti-5553 alloy, could occur during the hot compression in α + β field [16]. Interestingly, the dynamic α→β transformation of β titanium alloys during the sub-transus themalmechanical processing has also be been reported in Ti-10V-2Fe-3Al alloy, Ti-6Cr-5Mo-5V-4Al alloy, and Ti-5Al-2Sn-2Zr-4Mo-4Cr alloy [15,17,18]. Since the large-scale billet could not be cooled to room temperature at the high cooling rate that is the same as the water-quenching of the small-scale specimen, it is hard to characterize the microstructure of the billet that is immediately after α + β field forging. Zhao et al. found that the dynamic α→β transformation occurs during the α + β isothermal compression of Ti-5553 alloy at the compression strain rate range from 0.001/s to 0.1/s [19]. As a result, the volume fraction of the primary α phase decreases after the α + β isothermal compression. In addition, our recent work revealed that the primary α phase of Ti-5553 alloy completely disappears after the high strain rate (ε = 1000/s) compression at room temperature [20]. Therefore, we infer that the dynamic α→β transformation might occur during the α + β forging of the large-scale Ti-5553 alloy billet. In addition, the high pressure and the related stress concentration of the α + β forging may also lead to the α→ω→β transformation in the sub-transus forging [21,22].

In summary, the temperature rise and the dynamic α→β transformation, which is caused by the high strain rate forging, contribute to the formation of the incomplete recrystallization β region and the abnormal β-grain growth. Furthermore, we will investigate the role of forging strain rate in β grain size inhomogeneity of large-scale Ti-5Al-5V-5Mo-3Cr alloy billet by using the hydrostatic press.

## 5. Conclusions

In this paper, we conducted a series of multi-axial forging with 62 forging cycles on large-scale Ti-5553 billet and investigated the β grain size evolution of billet during the whole α + β forging series. The following conclusion could be drawn.


It is impossible to fabricate large-scale homogeneous Ti-5553 alloy billet with fine and equiaxial β grain by multi-cycle α + β forging with a high deformation rate.The inappropriate forging parameters, especially the deformation rate, might lead to the incomplete dynamic recrystallization of the β phase and abnormal grain growth of β phase during the multi-cycle and multi-axial forging of the large-scale Ti-5553 alloy billet.After performing the solution treatment at 785 °C and the 5 min ageing treatment at 785 °C, the β grain size and β grain shape of Ti-5553 alloy forging billet could be characterized by an optical microscope.


## Figures and Tables

**Figure 1 materials-16-01692-f001:**
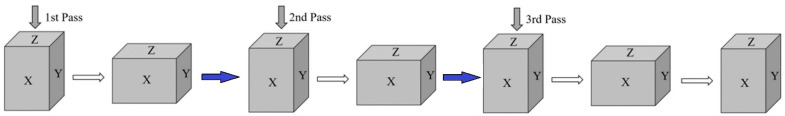
Process flow of β field forging (for each forging cycle).

**Figure 2 materials-16-01692-f002:**
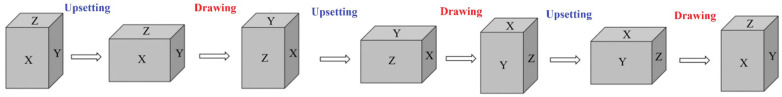
Process flow of α + β field forging (for each forging cycle).

**Figure 3 materials-16-01692-f003:**
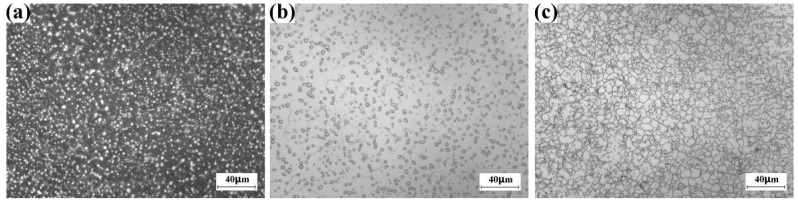
Optical images of large-scale Ti-5553 alloy billet after the 14th cycle forging. (**a**) Forged. (**b**) Solution condition. (**c**) Solution-ageing condition.

**Figure 4 materials-16-01692-f004:**
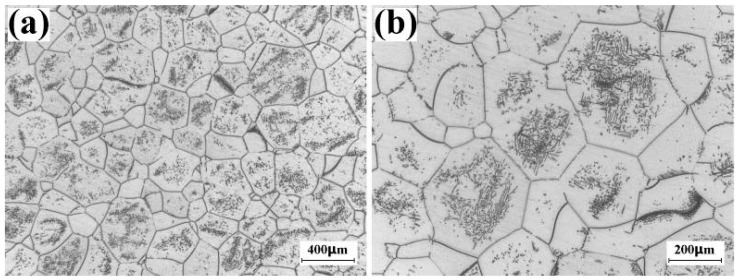
Ex situ investigation on the microstructure of large-scale Ti-5553 alloy billet before the multi-cycle and multi-axial forging in α + β field. The Φ 10 mm × 10 mm specimen was cut from the billet forged in β field, solutionized at 900 °C for 120 min following water-quenching and finally aged at 785 °C for 120 min following water-quenching. (**a**) Low-magnificaion optical image showing the grain size of β grain before the sub-transus forging. (**b**) High-magnification of optical image showing the intragranular α precipitation and grain-boundary α precipitation.

**Figure 5 materials-16-01692-f005:**
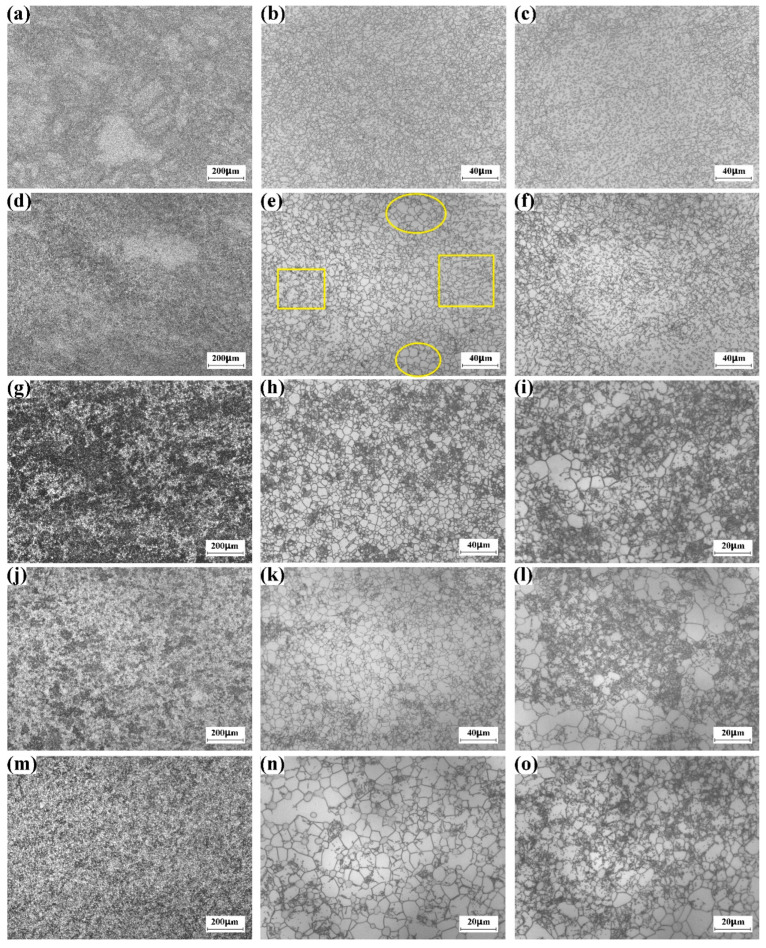
β grain size refinement of large-scale Ti-5553 alloy billet during multi-cycle multi-axial forging at 785 °C. (**a**–**c**) 6 cycles, (**d**–**f**) 14 cycles, (**g**–**i**) 22 cycles, (**j**–**l**) 38 cycles, (**m**–**o**) 44 cycles. In subfigure (**e**), the complete recrystallization region and the incomplete recrystallization region are marked by the yellow circles and the yellow rectangles, respectively.

**Figure 6 materials-16-01692-f006:**
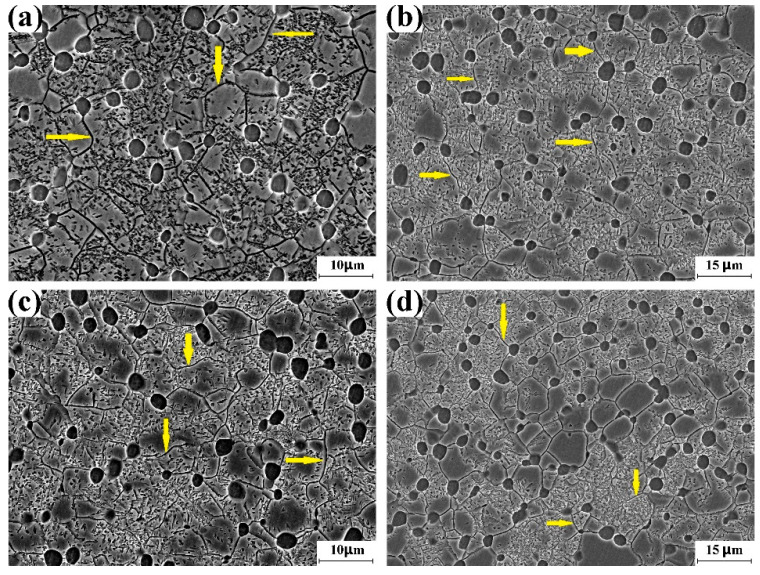
SEM image for incomplete recrystallization region. (**a**) 14th cycle, (**b**) 22nd cycle, (**c**) 38th cycle, (**d**) 44th cycle. β grain boundary of the unrecrystallized β phase is marked by the yellow arrows.

**Figure 7 materials-16-01692-f007:**
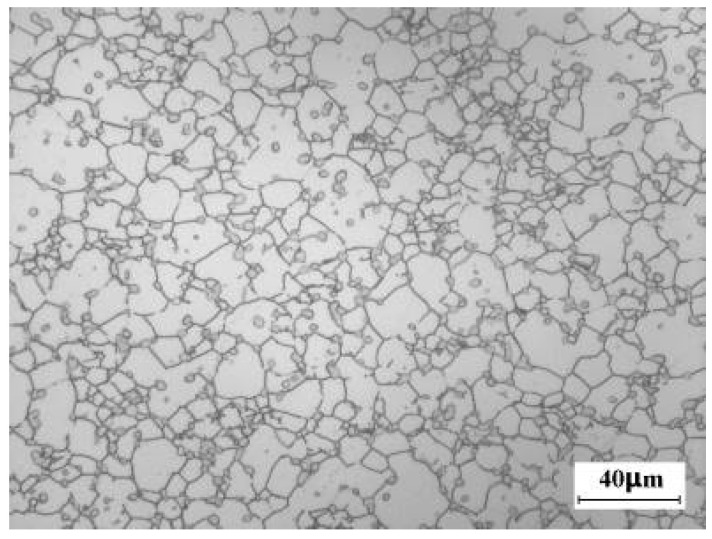
Eliminating the incomplete recrystallization region of large-scale Ti-5553 alloy billet by isothermal compression at 785 °C.

**Figure 8 materials-16-01692-f008:**
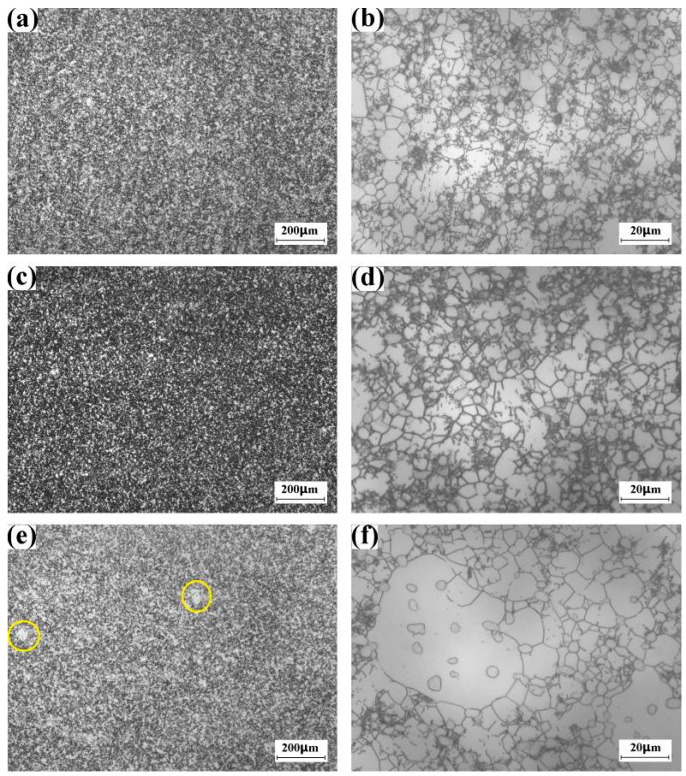
Abnormal grain growth of β phase for large-scale Ti-5553 alloy billet during multi-cycle multi-axial forging at 785 °C. (**a**,**b**) 50 cycles, (**c**,**d**) 56 cycles, (**e**,**f**) 62 cycles. In subfigure (**e**), the coarse β grain that is casued by abnormal grain growth is marked by the yellow circles.

**Figure 9 materials-16-01692-f009:**
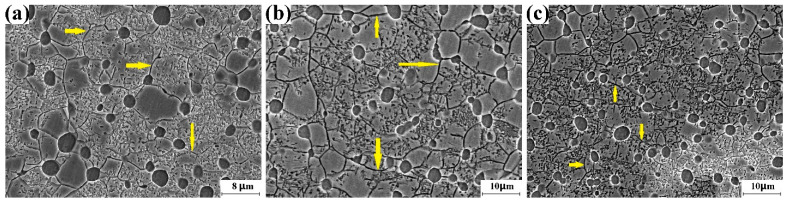
SEM image for incomplete recrystallization region. (**a**) 50th cycle, (**b**) 56th cycle, (**c**) 62nd cycle. β grain boundary of the unrecrystallized β phase is marked by yellow arrow.

**Figure 10 materials-16-01692-f010:**
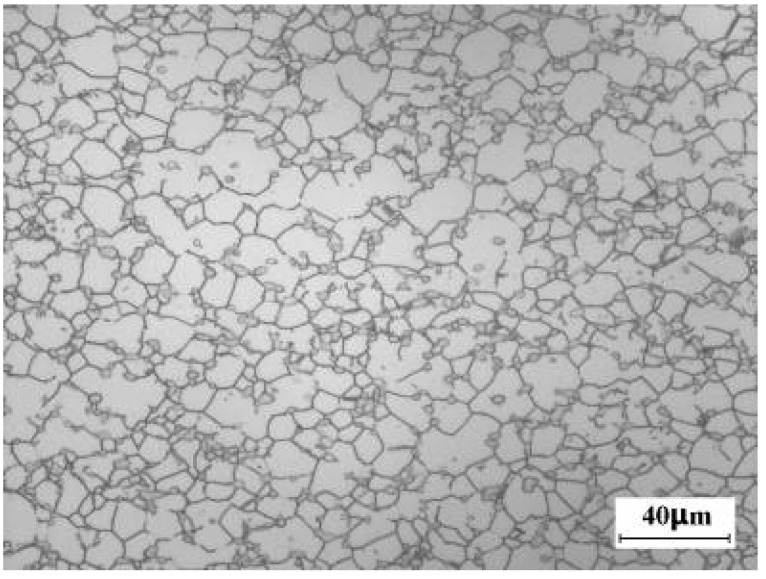
Eliminating the abnormally grown β grain of large-scale Ti-5553 alloy billet by isothermal compression at 785 °C.

**Table 1 materials-16-01692-t001:** Tensile performance of Ti-5553 alloy.

Microstructure	Average Grain Size (μm)	Yield Strength (MPa)	Tensile Elongation
β-annealed	200	900–1500 [2,3]	5–9% [2,3]
Bimodal	20	900–1400 [2,3]	12–20% [2,3]

**Table 2 materials-16-01692-t002:** Chemical composition of the Ti-5553 alloy ingot.

Elements	Al	V	Mo	Cr	O	Ti
**Mass Percent (wt. %)**	4.9	5.0	5.0	3.0	0.05	Bal.

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
