# Peer review of "β Grain Size Inhomogeneity of Large Scale Ti-5Al-5V-5Mo-3Cr Alloy Bulk after Multi-Cycle and Multi-Axial Forging in α + β Field"

_materials, 2023, doi:10.3390/ma16041692_

Round 1

Reviewer 1 Report

This is a very interesting work about the properties of titanium alloys. The paper is well written and merits to be published. Before it, I do have a few recommendations aimed to improve the manuscript.

1.       I recommend to comment in the introduction that the high-pressure phase stability and elastic properties of titanium alloys has been the focus of recent studies: citing D Smith et al 2017 J. Phys.: Condens. Matter 29 155401 and MacLeod S G, et al 2012 Phys. Rev. B 85 224202.

2.       Page 2 line 57. Is it correct Ti-5555?

3.       Page 3 line 109. There is a space missing in “ofβ phase”

4.       Same page, line 119, describe better “water-quenching”

5.       Provide specifications of ZEISS-Stemi2000c optical microscope.

6.       Consider adding a table with name and composition of alloys.

7.       Specify the source of samples.

8.       figure 5 (f) should be Figure 5(f). Check the naming of all figures.

Author Response

  1. I recommend to comment in the introduction that the high-pressure phase stability and elastic properties of titanium alloys has been the focus of recent studies: citing D Smith et al 2017 J. Phys.: Condens. Matter 29 155401 and MacLeod S G, et al 2012 Phys. Rev. B 85 224202.

The paper mentioned above has been cited in section 4.2, which is helpful in explaining the dynamic transformation of Ti-5553 alloy during sub-transus forging.

  1. Page 2 line 57. Is it correct Ti-5555?

It is not correct. The word “Ti-5555”has been revised as “Ti-5553”.

  1. Page 3 line 109. There is a space missing in “ofβ phase”

A space has been add between “of” and “β”.

  1. Same page, line 119, describe better “water-quenching”.

In the revised paper, we have described the water-quenching treatment in detail in section 2.1 as “After finishing each cycle of β-field forging, the billet was quenched into the water tank with the dimension of 2500mm×1500mm×1200mm.”

In section 2.2.1, the sentence “Before starting the α+β forging the billet was annealed at 900℃ for 120 minutes following water-quenching, and was then annealed at 785℃ for 120 minutes.” as has been revised as “Before starting the α+β forging the billet was annealed at 900℃ for 120 minutes and was quenched into the water tank, and was then annealed at 785℃ for 120 minutes.”

  1. Provide specifications of ZEISS-Stemi2000c optical microscope.

We are sorry of mistyping the type of the optical microscope. In the present work, Zeiss Axio Vert A1 optical microscope was used to investigate the microstructure of Ti-5553 alloy. The maximum amplification of the optical microscope is 1000.

  1. Consider adding a table with name and composition of alloys.

The table with name and composition Ti-5553 alloy has been added in section 2.1.

  1. Specify the source of samples.

In the revised paper, the source of metallographic samples has been specified in section 2.2.2 and section 2.3.2.

  1. figure 5 (f) should be Figure 5(f). Check the naming of all figures.

We have checked the naming of all figures.

Reviewer 2 Report

Dear Authors,

Please find below my comments/observations regarding your manuscript:

1. Comparing to the Introduction part which is very clear written, the experimental program is unclear described. So many stages with no strategy described; a reference is made to Ref [13] to read the strategy there; yes, but with such a bushy experimental program, the strategy considered should be clarified here as well, without a reference elsewhere; why first β-field forging and then α+β-field forging? In generally, is understandable from the introduction, but here for this particular case, it should be also explained;

2. The reasons for choosing such a program are not explained, the reasons for a specific sequence of operations (many, twisted and unexplained), the reasons for the selection of parameters, why certain temperatures, how the maintenance times were calculated, how the cooling media were determined are not explained; why some treatments are repeated? for example, an annealing is done at 780°C then the same annealing at 700°C; why? it is not explained. Please revise the experimental part.

Only after reading the entire manuscript, with the confusingly described experimental part and insufficiently proven results through inconclusive images, can one finally understand the strategy of the experiments from the discussions. Therefore, I suggest to relocate the strategy from discussion part in the beginning, in order to understand ab-initio what was the strategy of the work. The other comments/discussions from #4 are clear and well explained.

3. For the β-field forging no strain rate is indicated. In what direction are performed the multi-forging experiments, considering the 3D dimension of the samples? I suggest to add some schemas with 3D samples that can indicate the applied forging directions, and the exactly sampling area.

4. p.4, line 137: “The strain rate of the isothermal compression was 0.1/s. The isothermal compression strain was 0.69.    They refer to the same thing – strain rate, but with different values.

5. In my opinion, for the exemplification of the microstructure evolution during the forging process, the OM images are not enough. I suggest to add higher magnification using SEM images. And specifying the shooting direction for each image comparative to forging direction.

6. In 3.1#, for the Figure 2, the text indicates that: “In summary, we conclude that the microstructure of the cylinder should be the typical lamellar microstructure “. But the figure 2 doesn.t reveal any lamellar morphology, but an equiaxial one. Please, revise the provided commentaries.

7. Please enlarge the figures 3-a and 3-b; they look to be similar, even if the comments from the text indicate different/modified morphologies. The same for figures 3-e and 3-f.

8. p.6, 3.2.2#: “The complete recrystallization region and the incomplete recrystallization region could be observed “. The complete recrystallization region versus the incomplete recrystallization region must be indicated with proper arrows on the figures. In plus, these regions are very difficult to identify on OM images. Please provide SEM images with clear morphology particularities, that is much larger magnifications. The same for the higher forging cycles.

Author Response

The notes to reviewer 2 are in the attachment.

Reviewer 3 Report

The manuscript presents an important manufacturing process to obtain a Ti-5553 alloy with improved characteristics to be used in the avionics industry.

Some observations to increase the quality of the manuscript:

1. In the introduction section, please design a table in which you can provide the alloy's grain size and mechanical properties based on the literature study described in chapter 1.

2. In the Materials and Methods section and within the manuscript, please give the producer's name for the alloys and substances used and also for some measuring devices that were forgotten.

Author Response

  1. In the introduction section, please design a table in which you can provide the alloy's grain size and mechanical properties based on the literature study described in chapter 1.

The table regarding average grain size and tensile performance of Ti-5553 alloy has been added in chapter 1 of the revised paper.

  1. In the Materials and Methods section and within the manuscript, please give the producer's name for the alloys and substances used and also for some measuring devices that were forgotten.

Substances used for melting the ingot of Ti-5553 alloy were supplied by Northwestern Institute for Non-ferrous Metal Research. The maximum melting weight of the consumable electrode vacuum furnace is 1000Kg. These details have been added in the revised paper.

Round 2

Reviewer 2 Report

No more comments. Thank you.